# Simulation of the Attrition of Recycled Concrete Aggregates during Concrete Mixing

**DOI:** 10.3390/ma14113007

**Published:** 2021-06-01

**Authors:** Jaime Moreno-Juez, Luís Marcelo Tavares, Riccardo Artoni, Rodrigo M. de Carvalho, Emerson Reikdal da Cunha, Bogdan Cazacliu

**Affiliations:** 1Basque Research and Technology Alliance (BRTA), Astondo Bidea, Edificio 700, Parque Tecnológico de Bizkaia, 48160 Derio, Spain; jaime.moreno@tecnalia.com; 2MAST-GPEM, University Gustave Eiffel, IFSTTAR, 44344 Bouguenais, France; bogdan.cazacliu@univ-eiffel.fr; 3Department of Metallurgical and Materials Engineering, Universidade Federal do Rio de Janeiro, COPPE/UFRJ, Rio de Janeiro 21941-901, RJ, Brazil; tavares@metalmat.ufrj.br (L.M.T.); rodrigo@metalmat.ufrj.br (R.M.d.C.); emersonreikdal@gmail.com (E.R.d.C.)

**Keywords:** attrition, concrete mixing, discrete element method, recycled concrete aggregates

## Abstract

Concrete mixing can lead to mechanical degradation of aggregates, particularly when dealing with recycled concrete aggregates. In this work, the attrition of such materials during mixing is studied by means of experiments and simulations. The effect of the presence of fines, water addition, flow configuration of the mixer (co- or counter-current) and impeller frequency is discussed. Experiments were performed in a laboratory Eirich mixer. Discrete element numerical simulations (DEM) were performed on the same geometry by mimicking the behaviour of the material and, in particular, the cohesion induced by water and the cement paste using either Hertz–Mindlin or Hertz–Mindlin with Johnson–Kendall–Roberts (JKR) contact laws. The combination of the collision energy spectra extracted from the DEM simulations and an attrition model allowed the prediction of the mass loss due to attrition in 1-min experiments. Semi-quantitative agreement was observed between experiments and simulations, with a mean relative error of 26.4%. These showed that higher mass losses resulted from operation at the highest impeller speeds, co-current operation, and also with the wet aggregate. Mixing of the agglomerate in the concrete mix resulted in a significant reduction in attrition when compared to mixing aggregates alone. With further validation, the proposed simulation approach can become a valuable tool in the optimization of mixing by allowing the effects of material, machine and process variables to be studied on the mass loss due to attrition.

## 1. Introduction

Granular materials, such as aggregates, can be subject to changes in particle size distribution during the operations of transport, handling and mixing [1,2]. Such mechanical degradation can occur through body or surface breakage mechanisms [3]. Surface breakage, that is, surface wear, abrasion or attrition, can be a major concern for process control and downstream applications. For example, in concrete mixing, attrition may reduce the average particle size and increase the proportion of fines, and also reduce the angularity of particles. This has the effect of changing the formulation within the mixer, resulting in a more problematic control of the mixing process, reduced workability and the need for adding more water to obtain the required consistency. The reduction in the angularity of particles as a result of attrition, may, in turn, also have deleterious implications to the mechanical properties of the final concrete [4].

The societal path towards a circular economy in construction has in recent years pushed the problem of recycling into concrete. Recycled concrete aggregates (RCA) are a multicomponent mixture of mortar and natural aggregates. Their incorporation into new concrete has focused new light on the problem of mechanical degradation and attrition during mixing, which is more critical for this type of aggregate in comparison to the natural material. From this perspective, [5] studied the degradation of RCA experimentally for a concrete formulation in a pan mixer and identified responses that were particular to this type of aggregate. It was observed that the evolution of mass loss due to surface breakage with time was not linear and that the rate of attrition decreased with time. This can be explained by the fact that particle wear is sensitive to particle shape so that when attrition proceeds, particles become rounder, and their attrition rate decreases. It can also be explained by the fact that wear removes the mortar preferentially on the surface of the particles, making the resulting coarse particles less amenable to additional surface breakage. The second important point observed by Moreno-Juez et al. [5] is that there seems to be a discontinuity of behaviour after water addition, with the wet mixing phase appearing more aggressive than the dry one. There is, therefore, a need to update the techniques for characterizing attrition during the mixing of aggregates, and numerical techniques are a natural choice.

Important advances in the understanding of degradation and size reduction have been made possible with the wide application of the Discrete Element Method (DEM) [6,7]. The method allows describing the interactions amongst particles and between them and the machine [8,9,10], being amenable to be used in association with proper descriptions of breakage in predicting comminution and degradation during handling [6,11].

In order to model size reduction with DEM, two main approaches exist. The first is to use a suitable discretization of particles, where the grains are bonded together by breakable cohesive links [12,13,14,15], which may be broken progressively as a result of interactions between particles and between particles and the machine. This approach requires simulation of a large number of grains, which will increase with the fineness of the fragments that one wishes to simulate. With respect to attrition, such an approach becomes prohibitive due to a large number of grains and to the fine nature of the attrition products.

Another approach consists of simulating particles that are not allowed to break and thus do not evolve during the process. In this approach, the micromechanical information which can be extracted from DEM simulations is then used, often in association with experimental characterization, to estimate the outcome of the process. For instance, Han et al. [16] used the distribution of impact energies between particles obtained in simulations using DEM, combined with the experimental characterization of the effect of impact velocity on the intensity of breakage (determined with single particle impact tests), to estimate the extent of attrition of salt in a pneumatic conveying process. In a similar way, Ahmadian et al. [17] studied granule breakage in a rotary drum mixer by coupling DEM and single-particle impact tests. Attrition data obtained from controlled bulk shear experiments have been used by Hare et al. [18,19] in association with average stresses acting on particles obtained from numerical simulations to study attrition in agitated beds of pharmaceutical powders. Recently, collision energy spectra, along with an attrition model, have been used to predict the attrition of iron ore pellets during tumbling in a drum and sieve shaking, with good agreement between experiments and simulations [20].

Considering these studies, it seems that this second approach has a great, but yet unrealized, potential to assist in understanding the attrition of aggregates during mixing. After proper validation, this approach would be useful to predict RCA degradation during concrete mixing and thus avoid poor formulations and the consequent losses of physical and mechanical properties of new concrete while guaranteeing a good mixing response.

The present work combines experiments and DEM simulations to investigate attrition of recycled concrete aggregates during mixing in a laboratory-scale Eirich mixer operating under several different conditions.

## 2. Materials and Methods

### 2.1. Mixer Type

The tests were conducted in a 5-L laboratory intensive pan mixer (Eirich Gmbh, Hardheim, Germany, shown in Figure 1). Such a mixer is characterised by one impeller, one scraper, and an inclined rotating vessel that typically runs between 45 and 90 rpm, whereas the impeller can turn at frequencies that vary from 50 to 700 rpm. The mixer can be operated in two different configurations depending on the sense of rotation of the vessel and agitator: a co-current (CO) configuration in which both the vessel and agitator rotate in the same direction (clockwise) and a counter-current (CC) configuration in which the vessel rotates clockwise and the impeller rotates counter-clockwise.

In this study, the mixer was operated in co-current and in counter-current configurations at three impeller speeds (150, 300 and 500 rpm). The vessel rotation frequency was maintained constant at 45 rpm in all cases.

### 2.2. Materials

This study focused on the attrition of coarse aggregates in a granular paste. For this reason, a reference concrete mixture was designed, composed of 10–14 mm aggregate, natural 0–2.5 mm silico-calcareous sand from Lafarge Granulats (Cheviré, France), cement CEMI 52.5 from Lafarge Ciments (St. Pierre La Cour, France), and water. A detailed discussion of the experiments and a particular focus on the effect of the nature of the aggregates on the attrition phenomenon is given elsewhere [5]. The present work focused on the comparison between experiments and numerical simulations so that only results for one type of aggregate, recycled concrete aggregate (RCA), coming from the Gonesse Recycling Centre in France, are presented. These aggregates were composed of 99% recycled concrete and 1% of inert materials. Recycled concrete is a heterogeneous material mainly composed of natural aggregates (crushed rock) and residual from the mortar paste.

The properties of both the RCA and sand employed are presented in Table 1. The water absorption (specific amount of water absorbed in 24 h by the grains) is an indirect measure of the mortar content. The RCA used in this study was characterized by a mortar content between 5 and 10%.

In order to better understand the effect of the cement paste on the attrition of aggregates, three formulations were prepared (Table 2):Coarse aggregates, dry (AD): only the 10–14 mm RCA;Coarse aggregates, wet (AW): a mixture of the 10–14 mm RCA and water, initially mixed by hand, with the water amount chosen to slightly cover the aggregates when poured into the mixer;Concrete, wet (CW): reference concrete formulation.

### 2.3. Experimental Procedure and Mass Loss Estimation

The experimental procedure was as follows:The coarse aggregates were divided into equal samples with a sample splitter following the EN 932-2 standard [21]. This method for reducing laboratory samples allows obtaining statistically equal samples in terms of properties and characteristics;The materials, either AD or AW, were added to the mixer.For the CW formulation, the cement paste was first prepared in the mixer by mixing all the components (sand, cement and water) for 60 s at 500 rpm. Then the coarse aggregates were added to the mixer and gently incorporated into the paste by hand.The materials were mixed for 60 s at the selected frequency of rotation;The contents of the mixer were carefully retrieved;In order to isolate the coarse aggregates remaining after mixing, the materials were sieved under water on a 2.5 mm sieve; the recovered coarse aggregate (i.e., +2.5 mm) was dried in an oven (Memmert GmbH, Schwabach, Germany) at 70 °C for at least 72 h;The coarse aggregates were weighed after drying.

In order to characterize the attrition of coarse aggregates during mixing of the concrete formulation, the relative amount of aggregates falling below a reference particle size (2.5 mm) was chosen as a relevant parameter:(1)$$\Delta M=\frac{M_{initial}-M_{end}}{M_{initial}}$$
where *M_initial_* is the (initial) mass of aggregates larger than 2.5 mm in the feed sample (determined by oven drying a reference sample) and *M_end_* is the corresponding mass at the end of the test. The threshold value of 2.5 mm was chosen because it was the upper limit of the sand fraction used. A full particle size distribution analysis with a photographic method was performed on several samples [5] in order to validate this choice: such analyses confirmed that mixing induced an attrition of coarse aggregates with a strong separation between the characteristic sizes of fragments and supported the choice of the size threshold for quantifying attrition, which is characteristic of surface breakage [20]. The initial water content of materials was measured and taken into account for the calculation of mass loss.

### 2.4. Single Particle Impact Experiments

Several lots containing approximately 200 g of 10–14 mm RCA each (around 70 particles) were initially prepared. Each lot was weighed as a whole, and then particles were dropped individually against a thick steel plate. Drop heights equal to 1.02, 2.05 and 3.07 m were used in the experiments, and only one impact per particle was performed. Given that no particles lost significant debris at each individual impact event, which would characterize volume breakage [20], the lot was weighed again after the impact in order to determine the average percentual mass loss per impact. This information was the basis for the calibration of the attrition model presented in Section 3.2.

## 3. Numerical method

### 3.1. Simulation Setup

Numerical simulations using the Discrete Element Method were conducted using EDEM^®^ 2.7 from Altair EDEM (Edinburgh, UK).

Data initially used to simulate the behaviour of particles in the Eirich mixer were derived from a rock that is used for production of aggregates, which was also used in simulations of a vertical shaft impact crusher [22]. These included the static and rolling friction coefficients, the coefficient of restitution, the physical properties, etc. The data were considered sufficiently reliable so that they could be used to perform simulations that mimic the reality in regard to the movement of the particles and the energy transfer in the mixer. It is known that aggregate particles, either recycled or natural, are not spherical. In the simulations, they were modelled as two partially superimposed (clumped) spheres, resulting in particles with an aspect ratio of approximately 0.7 (Figure 2). Simulated particles presented sizes ranging from 10 to 14 mm (Table 1), mimicking the measured size distribution with an average size of 12.2 mm, and 54% passing 12 mm. Each simulation involved a total of about 2200 aggregate particles.

Simulations were performed using two contact models: no-slip Hertz–Mindlin [23] and Hertz–Mindlin with JKR Cohesion (Johnson–Kendall–Roberts) [24] from the library of models available in EDEM 2.7. The first one was used to represent simulations of non-cohesive systems, namely mixing of the dry aggregate (AD). The second model was originally proposed to allow for the simulation of Van der Waals forces [25], which influence the flow behaviour of fine and dry powders. However, it also allows representing the cohesive nature of both fine particles and wet materials, being capable of describing the influence of moisture content on the mass flow of larger-scale materials, such as iron ore or wet grains [26,27]. This model was used to simulate the mixing of the wet formulations, that is, AW and CW (Table 2).

As mentioned, some of the data from previous studies [22] only served as initial estimates for the calibration of contact parameters. Bench-scale tests were performed to determine the aggregate angle of repose and the value of the angle used as a reference for the simulations. The simulations were used to ensure that the behaviour of the simulated particles was as close as possible to that of the real particles. From the reference values of static and dynamic friction, the simulation was set up to predict the same value as the angle formed in the laboratory angle of the repose test. As described, mixing tests using the Eirich mixer included not only the recycled concrete aggregate particles but also sand, cement and water. Unfortunately, simulation of the entire charge using DEM would not be practical, so that it was decided to include only the coarse aggregate particles (Figure 3).

A fine tuning of the contact parameters was carried out by trial-and-error by comparing the pattern of the charge motion in the mixer, observed in videos of similar mixers operating under comparable conditions. A summary of the material and contact parameters selected is presented in Table 3 and Table 4.

Simulations were then conducted to simulate the operation of the mixer under a variety of conditions, namely mode of operation (co-current or counter-current), impeller speed (150, 300 and 500 rpm), using both cohesive and non-cohesive contact models. A time step of 6.5 × 10^−7^ s was chosen in the simulations, which is equivalent to 1% of the Rayleigh time. Values of time step as short as this are required in such a system given not only the high speed achieved by particles but also the collision energy logging required to generate the energy spectra and ensure that more than 97% of contacts achieve a maximum overlap of 0.3% of sphere radius. This is in accordance with the work by Marigo and Stitt [28]. In order to collect enough information for computing the collision energy spectra, 120 s of the operation of the mixer was simulated. Data belonging to the initial transient period required to attain a steady state (determined by analysis of the collision spectra on selected time intervals) were not considered in the computation of the energy spectra. In addition to the overall collision spectra, the history of collisions of individual aggregate particles was logged. Particles remained unbroken in the simulations.

In order to generate the collision energy spectra, after the mixer reached a steady state, the collision data were extracted, that is, the frequency and magnitude of energy dissipated in each collision involving two elements, namely particle–particle, particle–walls or particle–impeller. Then, each collision was placed into one of 1000 bins ranging from 10^−16^ to 10^1^ J following a logarithmic scale. Both normal and shear components were recorded, and their sum was used, given recent evidence of their joint contribution to surface breakage [20]. Each value of total collision energy loss was then split between the elements involved in the collision on the basis of Hertz elastic theory [29], assuming a perfectly elastic impact. As such, for a collision between a particle *p* and another body *q* (particle or wall), the fraction of collision energy available for damaging particle *p* (*e_p_*) is computed by:(2)$$e_{p}=\frac{\;Y_{q}/\left(1-\upsilon_{q}^{2}\right)}{Y_{p}/\left(1-\upsilon_{p}^{2}\right)+Y_{q}/\left(1-\upsilon_{q}^{2}\right)}$$
where *ν* and *Y* are defined in Table 3.

The total energy loss per particle is then converted into specific energy by dividing it by the average mass of each particle involved in that event, giving:(3)$$E=\frac{E_{loss}e_{p}}{m}$$
where *E_loss_* is the total energy loss in the collision, collected from DEM, and *m* is the mass of particle *p*.

### 3.2. Attrition Model

Considering that, for the mixing conditions and materials investigated in the experiments, no evidence existed of massive or volume breakage of coarse aggregates, only surface breakage or attrition, modelling was only focused on this latter mechanism. In a recent work, Cavalcanti et al. [20] proposed a modification of the model by Ghadiri and Zhang [30] through which the average percentage mass loss in each collision of iron ore pellets may be calculated by:(4)$$\bar{\xi }=100kdE$$
where *E* is the mass-specific energy loss of particle *p* that incorporates both the normal and the shear energy loss contributions and *d* is the representative size of the particles. The material-dependent parameter *k* in Equation (4) represents the amenability of material to surface breakage, which should be estimated from single-particle impact experiments [20]. In this form, the model becomes directly usable with data provided by DEM simulations.

The prediction of attrition for the aggregates in the mixer was possible by applying Equation (4) successively to each collision and particle, giving:(5)$$M_{end}=\overset{n}{\underset{i=1}{\sum }}m_{i}\;\overset{n_{i}}{\underset{j=1}{\prod }}\left(1-kd_{i}E_{i,j}\right)$$
where *m_i_* is the initial mass of each simulated particle, *n* is the total number of particles simulated, *n_i_* is the total number of collisions each particle suffered during the course of the simulations and *E_ij_* the specific energy loss for particle *i* at collision *j*. The percentage mass loss in the simulations was then obtained from Equation (1), recognizing that the initial mass of the particles was given by $M_{initial}=\sum m_{i}\;$.

The attrition model given by Equation (4) has been successfully validated for iron ore pellets, which present a nearly spherical shape [17]. It did not take into account the higher initial mass loss due to irregularity in shape, which has been known to occur in the case of RCA [5].

Given the fact that the ultimate application that is desired for the method is to predict the degradation of recycled aggregate during concrete mixing, it is evident that the presence of the interstitial paste makes collision phenomena more challenging both to model and measure. In order to account for the cushioning effect due to the presence of interstitial paste, as well as for the fact that deviations might exist between the parameter estimated from single-particle impact tests and the one prevailing during mixing of irregularly-shaped particles, Equation (4) has been modified through the introduction of a fitting parameter α, giving
(6)$$\bar{\xi }=100k\alpha dE$$

## 4. Results and Discussion

### 4.1. Single-Particle Impact Tests

The validity of the attrition model (Equation (4)) for aggregate particles was demonstrated by dropping aggregate particles, one by one, on a thick metal plate at different impact velocities. Figure 4 compares data for the RCA to that of a natural aggregate, as well as of these data to fitting to Equation (4). For the case of the RCA, the value of the attrition parameter $\bar{\xi }$ from fitting Equation (4) was 0.025 s^2^ m^−3^. The values of total energy loss were estimated from simulations with the no-slip Hertz–Mindlin model in DEM, dropping the particles one by one using material and contact parameters given in Table 3 and Table 4.

### 4.2. DEM Simulation Results

Snapshots from simulation results of the mixer operation are presented in Figure 5 and Figure 6 for the impeller frequencies of 150 and 500 rpm, respectively. At first, it is evident in the figures that impeller frequency, mode of operation and contact model had significant effects on both the motion pattern and the velocities of the aggregate particles in the simulations. A comparison of Figure 5 and Figure 6 shows that maximum particle velocities increased with increasing impeller frequency. When operating at higher impeller frequencies, particles close to the stirrer reached velocities up to 4 m s^−1^.

When the non-cohesive material was mixed, that is, when the Hertz–Mindlin contact model was used, the motion patterns responded to the mode of rotation and were relatively independent of impeller frequency. In the case of the counter-current (CC) mode, the material charge distributed, forming a nearly horizontal surface (left of Figure 5 and Figure 6). On the other hand, when the Hertz–Mindlin with JKR model was used, which is meant to mimic the condition in which water is added, the joint effect of the vessel and stirrer rotation was a lift of the charge, which then developed a surface profile that was nearly parallel to the vessel inclination. In this latter case, it was also evident that the charge became highly expanded vertically.

For both co-current (CO) and counter-current (CC) configurations, at lower stirrer speeds, particles closer to the walls of the vessel moved along the clockwise direction for both non-cohesive and cohesive environments (Figure 5). However, at high speeds, specifically in the counter-current 500 rpm case (Figure 6), particles collided with the back of the tip. In addition, particles were found to concentrate on the opposite side of the tip, and the effect of the rotating walls was less significant. This can be more clearly seen in Figure 7, which shows the top view of the vessel.

In the case of the co-current operation (Figure 7), higher particle velocities were observed, whereas the bed of aggregates was less expanded along the axis of the impeller, regardless of its frequency when compared to the counter-current mode.

A more direct analysis of the aggressiveness of the mixer, when operated under different conditions, was possible through the analysis of the collision energy spectra. Figure 8 displays the high-energy part of the collision energy spectra for simulations without cohesion, which emulates dry mode operation. It is clear, as one could expect, that the collision energy spectra widened when the impeller speed was increased. The flow configuration also had a strong influence on the shape of the collision spectra, with the co-current being responsible for higher frequencies of collisions in the intermediate and high-energy part of the spectrum for 300 and 500 rpm. Spectra were found to display power-law tails (low energies) decaying approximately with a −2 exponent.

In Figure 9, the collision energy spectra for simulations with cohesion (Hertz–Mindlin with JKR) are displayed. It is evident that the effect of impeller speed was analogous to that in the preceding case. On the other hand, some differences with respect to the effect of the flow configuration can be noticed. In the high-energy tail, the co-current configuration had a wider distribution, and this would likely result in more energy available for attrition. The frequencies of collisions of magnitude higher than about 1 J/kg per particle for the co-current cases were approximately 40% higher than those obtained for counter-current cases for rotation speeds of 300 and 500 rpm. In Figure 9, it is also evident that the introduction of cohesion forces through the JKR model had the effect of widening the collision energy spectrum in comparison to those in Figure 8.

### 4.3. Measurement and Simulation of Attrition

The effect of impeller frequency and flow configuration on the mass loss observed experimentally was analysed as follows. At first, results from tests involving only aggregates (AD and AW in Table 2) were analysed. The values of mass loss after 60 s of mixing for the dry and wet aggregates are displayed in Figure 10 for co- (CO) and counter-current (CC) configurations as a function of impeller speed. At first, it is evident that impeller speed was the most significant effect influencing mass loss due to attrition. A comparison of results for the different flow configurations demonstrated that the co-current configuration yielded more attrition for all impeller speeds studied. A less marked effect was associated with the presence of water, which was responsible for, in general, increasing the attrition of the RCA.

In order to understand the effect of the different conditions tested during mixing, predictions using the attrition model given by Equation (4) were compared to experiments. At first, only data on the dry (AD) and wet (AW) mixing of the coarse aggregates were considered, which were mimicked by the simulations using no-slip Hertz–Mindlin and Hertz–Mindlin with JKR, respectively, given the effect of water in creating cohesion of the aggregates. In this case, the value of $\bar{\xi }$ equal to 0.250 s^2^m^−3^ estimated from Figure 4 was used for the attrition amenability of the RCA. However, simulations demonstrated that the values of predicted mass loss were about five times higher than those shown in Figure 10. Given that the value of $\bar{\xi }$ estimated from single-particle impact tests represents the first event of mass loss, which is typically significantly higher, given the irregularity in particle shape, and also to the fact that a cushioning effect appeared due to fines that were generated that dissipate part of the energy in the collisions, it is recognized that the use of Equation (6), in which the calibration factor *α* appears, was necessary. As such, the value of α equal to 0.18 (18%) was used in simulations involving only aggregates (AD and AW).

Therefore, Figure 11 presents results from the combination of simulation data and the attrition model for the different formulations depicted in Table 2. Attention is first given to the results from the predictions using the coarse aggregate alone. The general semi-quantitative agreement with Figure 10 is clear, with the combination of the attrition model (Equation (6)) and the DEM simulations being able to properly account for the effect of impeller speed, as well as for mode of operation, if co-current or counter-current, with a mean relative error, given by the ratio between difference in estimates and the experimental value, equal to 29.0%. The effect of water addition increasing the attrition at speeds higher than 150 rpm in the experiments (Figure 10) was also observed in the simulations (Figure 11) but was only observed with the highest speed simulated (500 rpm).

Finally, both experimental and simulation results were analysed for the wet concrete formulation (Table 2). When comparing the attrition behaviour of the concrete formulation to those characterized by only coarse aggregates (Figure 10 and Figure 11), it is clear that the latter were more damaged when mixed alone. The presence of mortar between the aggregate particles had a protective and cushioning effect, which can be ascribed to the dissipation of the collision energy by the paste. Another point that may influence the result is the difference in batch volume (Table 2). It is reasonable to assume that attrition phenomena were more likely to occur near the impeller of the mixer, where shear was higher, and collisions were more frequent. The impeller was located in the lower half of the mixer so that the increase in batch volume would expose more material to a zone of lower shear, where attrition was not as significant.

Figure 11 also shows that attrition of CW also increased with the impeller frequency. Another interesting point is regarding the flow configuration of the mixer, in which the counter-current configuration seemed to be more aggressive for low impeller speeds, while the co-current one yielded more attrition for high impeller speeds. This behaviour was also observed in [5].

Application of the model to predict attrition in the concrete mixtures is even more challenging, given the cushioning effect provided by the fine aggregate and cement paste, which reduces attrition of the coarse aggregate. Given that particles responsible for the cushioning effect were not included in the simulations, the factor α in Equation (5) would need to be further modified. The value of 0.05 (5%) was used in the simulations shown in Figure 11, which presents the same general trend for those obtained with the wet aggregate (AW), only with a lower magnitude. As such, co-current operation predicted greater attrition of RCA. Such results were in agreement only with the experimental result at the impeller frequency of 500 rpm. Nevertheless, the mean relative error between experimental and simulation results for aggregate degradation in the concrete mixture was 21.3%. 

This work aimed to be a first step in the use of discrete methods for the understanding of attrition of recycled aggregates during concrete mixing. From this perspective, one of the main interrogations of the present work was to determine if discrete element model simulations of coarse grains with cohesion could be somewhat representative of the attrition behaviour of coarse aggregates in a concrete formulation. It is clear that the main trends of the attrition behaviour in the mixer were captured by the numerical simulation, too (effect of water addition, effect of the flow configuration). Nevertheless, there were some deviations, particularly for low impeller speeds, which, also considering the experiments performed on coarse aggregates alone, could be ascribed to the polydisperse nature of the concrete mixture. However, we can state that, even if it is necessary to employ some relatively crude approximations when attempting to model concrete mixing using the discrete element method, such numerical simulations can definitely be useful to understand the effect of process parameters on the attrition of particles in the mixer.

## 5. Conclusions

In this paper, we evaluated a numerical approach to predict the attrition of recycled concrete aggregates during mixing. The approach combined DEM simulations of the coarse fraction with an attrition model, which allows computing the rate of mass loss of the aggregates as a function of the energy dissipated by collisions, as obtained from the simulations. The effect of water or mortar on the motion of the aggregates in the mixer was mimicked by a cohesive contact law (JKR). The material-dependent propensity to attrition was included in the model through a parameter that was determined by single-particle impact tests. Reduction in attrition by cushioning due to fines was also considered through an empirical correction.

In the case of experiments involving only aggregates, it was found that the impeller speed was the most significant effect influencing mass loss due to attrition. A comparison of results from the different flow configurations demonstrated that the co-current configuration yielded more attrition for all impeller speeds studied and the counter-current. A less marked effect was observed in respect to the presence of water, which was responsible for, in general, increasing the attrition of the RCA. 

A comparison between experiments and simulations involving only aggregates demonstrated that the proposed approach, which combined DEM with the appropriate contact model and a correction factor for the attrition model to account for the cushioning effects, led to good semi-quantitative agreement between them, with mean relative errors of 29.0%. In particular, similar behaviour was found for the joint effect of impeller frequency, water addition and flow configuration. Indeed, simulations also showed that the main parameter controlling aggregate attrition was mixer impeller frequency, that the co-current mixing configuration was responsible for higher attrition, and that the difference between the intensity of attrition in co-current and counter-current operation increased with impeller frequency. The effect of water addition, which was described indirectly by the contribution of adhesion in the Hertz–Mindlin model with JKR, resulted in higher attrition for higher mixer speeds when compared to the mixing of the dry aggregate as observed in the experiments. 

Experimental data on attrition of aggregates during mixing in a concrete formulation showed similar trends but with a smaller number of fines generated by attrition. In the model, this configuration was simply mimicked by a modification of the cushioning factor, which is a quite strong simplification which, however, yielded reasonable results. Still, the mean relative error between predicted and measure mass loss due to attrition in 1-min mixing was 21.0%.

Due to the approximations made (neglecting fines, simple cohesive model), and also to the definition of the attrition model, the comparison between experiments and simulations was only semi-quantitative. In order to perform more quantitative comparisons, detailed experiments should be performed. For instance, the effect of impact angle and cushioning by fines could be analysed through experiments in which particles are dropped one by one against either an angled steel plate or a surface containing fines, in addition to the variation of the mass loss due to attrition with the number of impacts. The approach in the paper was also only limited to cases when only surface breakage occurred. On the other hand, the present approach already has the advantage of computational efficiency since valid predictions are obtained without the need to include the fines explicitly in the simulations. 

The simulation approach proposed herein, with a mean relative error of 26.4%, can already be used to compare the mass loss due to attrition as a function of material characteristics (composition and size) as well as machine and process variables (vessel rotational speed, angle, impeller geometry and speed, filling level, medium), being potentially useful also to investigate the scale-up of the operation.

## Figures and Tables

**Figure 1 materials-14-03007-f001:**
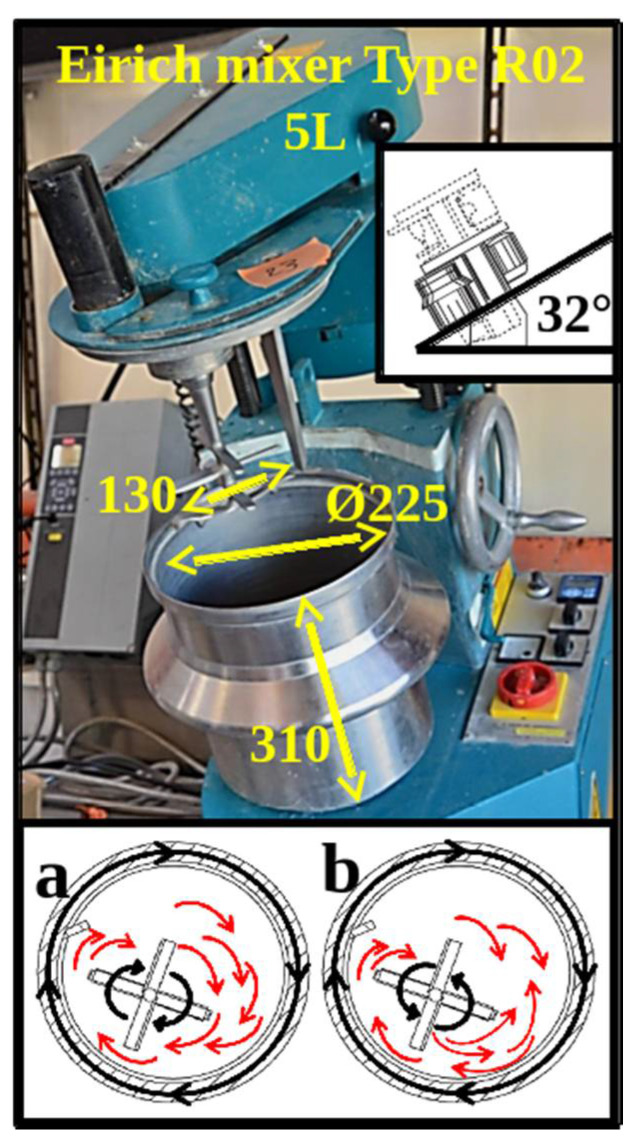
Laboratory mixer used in this study (a 5l intensive pan mixer, manufactured by Eirich Gmbh), illustrating the different modes of operation: co-current (**a**) and counter-current (**b**).

**Figure 2 materials-14-03007-f002:**
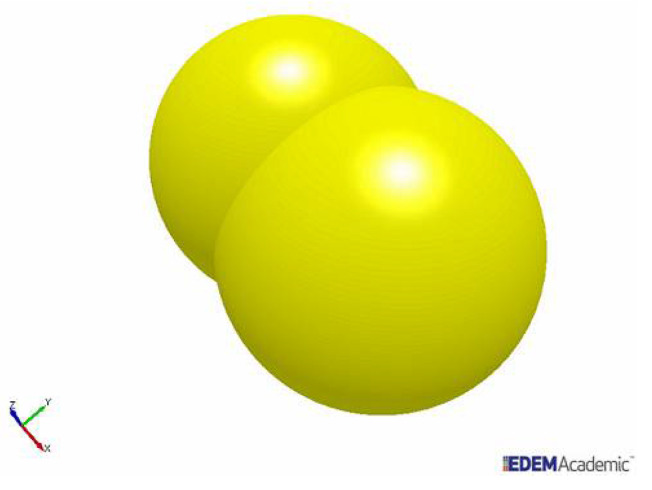
Model particle with an aspect ratio of 0.7 used in the simulations.

**Figure 3 materials-14-03007-f003:**
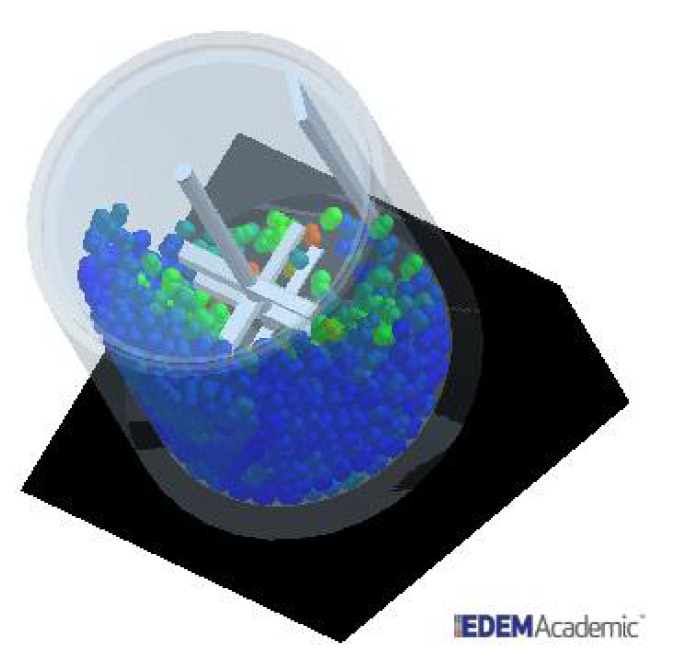
Snapshot of the simulation of the Eirich mixer using EDEM. The base of the device is inclined 32° in respect to the horizontal. Particles are coloured according to their velocity.

**Figure 4 materials-14-03007-f004:**
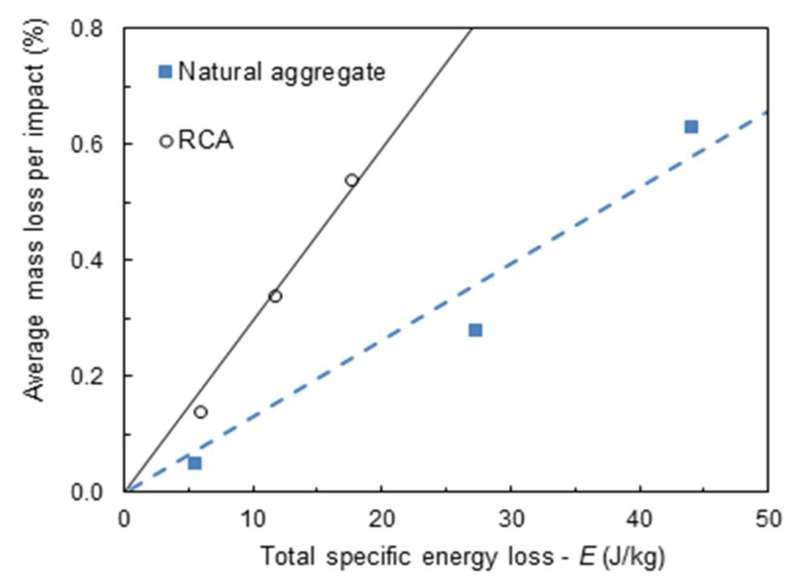
Comparison between measurements of average mass loss per impact and Equation (4) with the fitted value of $\bar{\xi }$ for the natural aggregate (19–22 mm) from the earlier work by Neves and Tavares [31] and to the recycled aggregate (RCA) from the present work (10–14 mm) ($\bar{\xi }$ = 0.0063 s^2^m^−3^ for natural aggregate and 0.0250 s^2^m^−3^ for RCA).

**Figure 5 materials-14-03007-f005:**
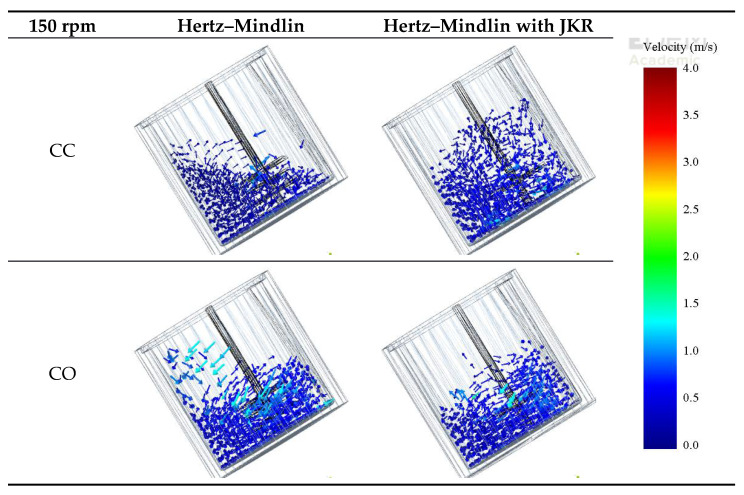
Comparison of simulated velocities of particles for the different contact models and modes of operation for an impeller frequency of 150 rpm.

**Figure 6 materials-14-03007-f006:**
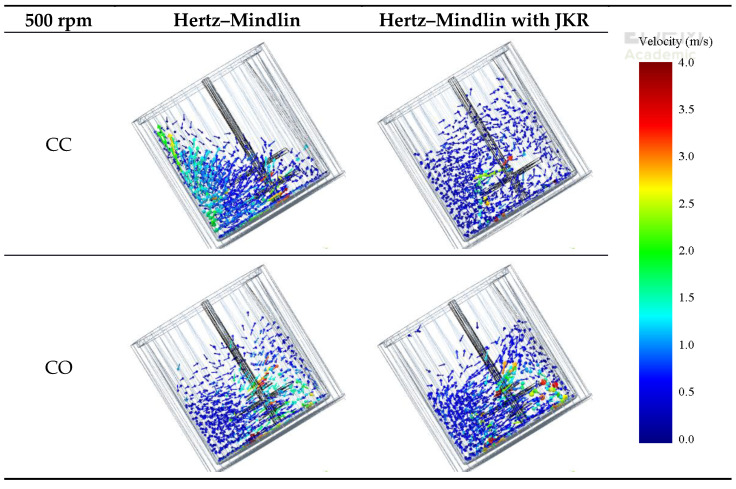
Comparison of simulated velocities of particles for the different contact models and modes of operation for an impeller frequency of 500 rpm.

**Figure 7 materials-14-03007-f007:**
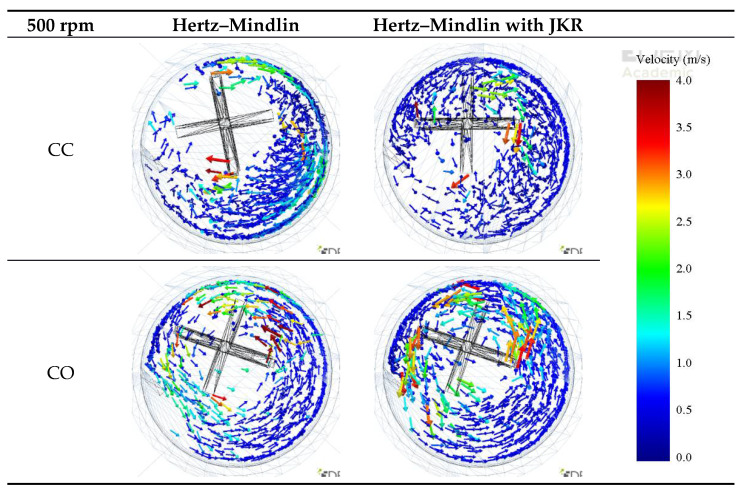
Top view of the comparing simulated velocities of particles for the different contact models and modes of operation for an impeller frequency of 500 rpm.

**Figure 8 materials-14-03007-f008:**
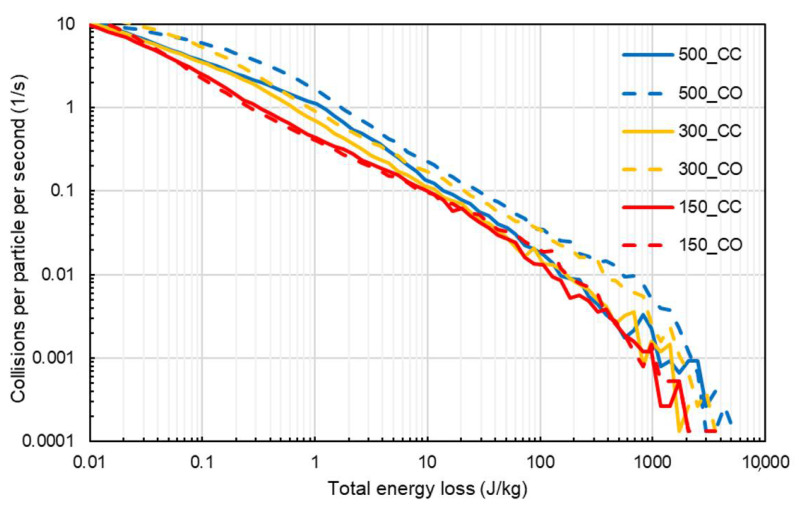
High-energy part of the collision energy spectra for simulations with different values of the impeller speed (in rpm), for counter-current (CC—solid lines) and co-current (CO—dashed lines) configuration, for simulations without cohesion (no-slip Hertz–Mindlin).

**Figure 9 materials-14-03007-f009:**
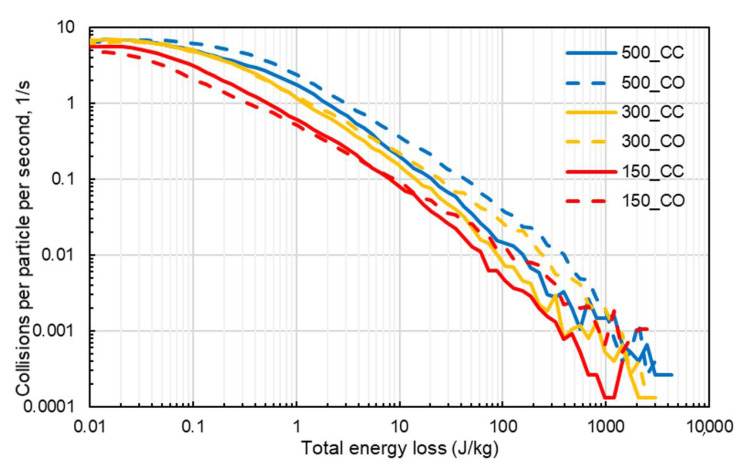
High-energy part of the collision energy spectra for simulations with different values of the impeller speed, for counter-current (solid lines) and co-current (dashed lines) configuration, for simulations with JKR cohesive model.

**Figure 10 materials-14-03007-f010:**
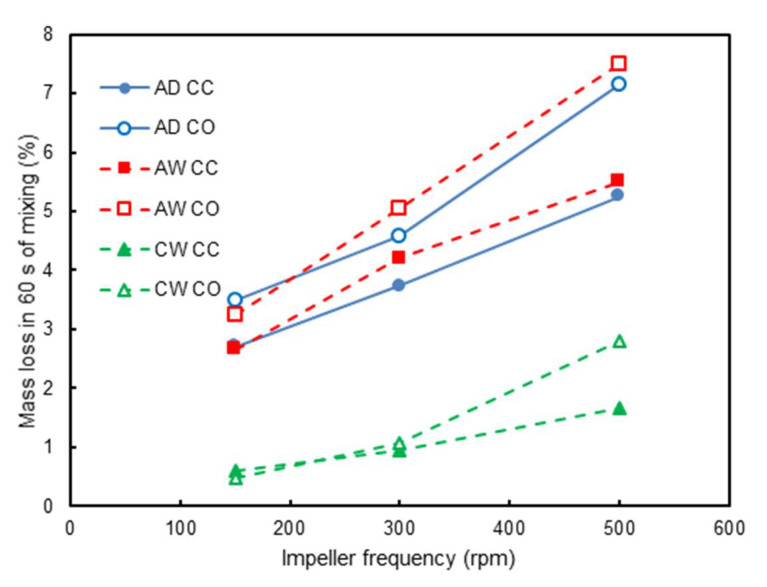
Experimental results on the effect of impeller frequency on the mass loss for tests with coarse aggregates only (AD, dry mixing, and AW, wet mixing) as well as a concrete formulation (CW), co- and counter-current configurations and 60 s of mixing.

**Figure 11 materials-14-03007-f011:**
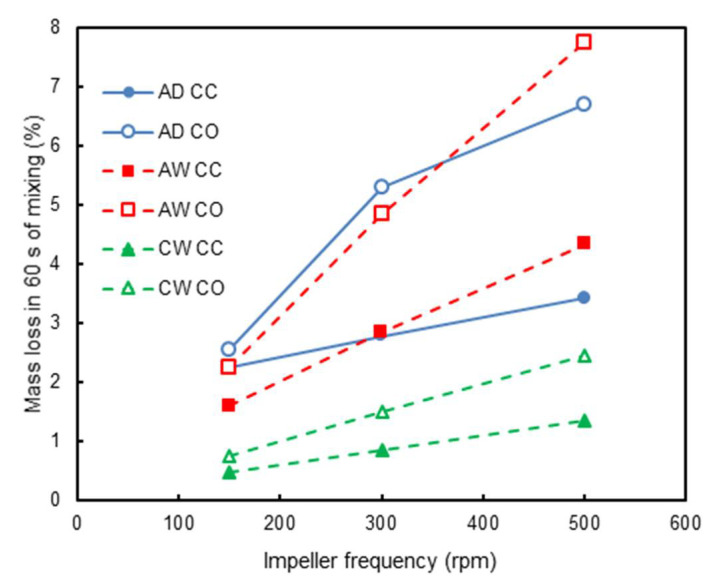
Simulations on the effect of impeller frequency on the predicted mass loss for tests with coarse aggregates only (AD, dry mixing, and AW, wet mixing), co- and counter-current configurations and 60 s of mixing, as well as for the wet concrete mixture (CW). Dry mixing was modelled by means of a non-slip Hertz–Mindlin contact law, while wet mixing is simulated by a cohesive contact model (Hertz–Mindlin with JKR). Mass loss is predicted using Equations (5) and (6).

**Table 1 materials-14-03007-t001:** Physical properties of sand and coarse aggregates.

Aggregate Type	Origin	Size (mm)	Density (kg/m^3^)	Water Absorption (%)
Recycled concrete aggregate (RCA)	*Gonesse* (France)	10–14	2290	5.1
Silico-calcareous sand	*Cheviré* (France)	0–2.5	2640	0.3

**Table 2 materials-14-03007-t002:** Formulations employed in this study.

Formulation	RCA, 10/14 mm (g)	Sand (g)	Cement (g)	Water (g)
Coarse aggregates, dry (AD)	4000	-	-	-
Coarse aggregates, wet (AW)	4000	-	-	1500
Concrete, wet (CW)	4000	4000	1400	950

**Table 3 materials-14-03007-t003:** Summary of material parameters.

Material	Poisson’s Ratio—*ν*	Young’s Modulus—*Y* (MPa)	Density (g/cm^3^)
Steel	0.30	182	7.7
Particles	0.30	60	2.5

**Table 4 materials-14-03007-t004:** Summary of contact parameters used in the simulations.

Property	Contact Type	
	Particle–Particle	Particle–Wall
Coefficient of static friction	0.35	0.28
Coefficient of rolling friction	0.35	0.25
Coefficient of restitution	0.40	0.35
JKR model cohesion parameter (J/m^2^)	10	0

## Data Availability

Data sharing is not applicable to this article.

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
