# Peer review of "Simulation of the Attrition of Recycled Concrete Aggregates during Concrete Mixing"

_materials, 2021, doi:10.3390/ma14113007_

Round 1
Reviewer 1 Report
The text and the content of the paper are clear and readable.
Remarks:
1. Please, write clearly in the paper whether the simulations are performed using models, i.e. no-slip HM and HM with JKR, which are implemented by the Authors (as e.g. own subroutine for material model) or are only available in EDEM 2.7 (as library code).
2. Reference no. 5 cannot be treated as basic for DEM! The Authors should broaden the discussion for this method.
The reviewer recommends the following literature:
- very basic and important paper:
Cundall, P. A. & Strack, O. D. L. (1979). A discrete numerical model for granular assemblies. Geotechnique, 29(1), 47-65. doi:10.1680/geot.1979.29.1.47 - alternative approach:
Beuth, L., Wieckowski, Z. & Vermeer, P. A. (2011). Solution of quasi-static large-strain problems by the material point method. International Journal for Numerical and Analytical Methods in Geomechanics, 35(13), 1451-1465. doi:10.1002/nag.965 - comparison of discrete models:
Rojek, J., Labra, C., Su, O. & Oñate, E. (2012). Comparative study of different discrete element models and evaluation of equivalent micromechanical parameters. International Journal of Solids and Structures, 49(13), 1497-1517. doi:10.1016/j.ijsolstr.2012.02.032
3. Line 256, page 7, should be; Eq. (4) - please, use the brackets.
Author Response
Response to Reviewer 1 Comments
The authors of this manuscript greatly appreciate the editor of the journal and the reviewers for the valuable feedback and comments provided. The authors have carefully considered all comments and suggestions forwarded by the reviewers. Accordingly, they are addressed in the revised manuscript.
The reviewers’ comments are listed below. Author replies are given under each comment (in red).
Point 1: Please, write clearly in the paper whether the simulations are performed using models, i.e. no-slip HM and HM with JKR, which are implemented by the Authors (as e.g. own subroutine for material model) or are only available in EDEM 2.7 (as library code).
Response 1: Indeed, all contact models were used as available in EDEM 2.7. This is clearly stated in the revise form of the manuscript.
Point 2: Reference no. 5 cannot be treated as basic for DEM! The Authors should broaden the discussion for this method.
Response 2: Acknowledge. A reference to the original Cundall and Strack (1979) has been included. An additional statement has been included with the aim to broaden the discussion for this method.
Point 3: The reviewer recommends the following literature:
Very basic and important paper:
Cundall, P. A. & Strack, O. D. L. (1979). A discrete numerical model for granular assemblies. Geotechnique, 29(1), 47-65. doi:10.1680/geot.1979.29.1.47
Alternative approach:
Beuth, L., Wieckowski, Z. & Vermeer, P. A. (2011). Solution of quasi-static large-strain problems by the material point method. International Journal for Numerical and Analytical Methods in Geomechanics, 35(13), 1451-1465. doi:10.1002/nag.965
Comparison of discrete models:
Rojek, J., Labra, C., Su, O. & Oñate, E. (2012). Comparative study of different discrete element models and evaluation of equivalent micromechanical parameters. International Journal of Solids and Structures, 49(13), 1497-1517. doi:10.1016/j.ijsolstr.2012.02.032
Response 3: The classical work by Cundall and Strack (1979) has been included, as suggested, as well as the last reference, which is certainly pertinent to the manuscript.
Point 4: Line 256, page 7, should be; Eq. (4) - please, use the brackets.
Response 4: Modified as requested.

Reviewer 2 Report
The manuscript developed DEM simulations to predict the attrition of RCA during mixing. The paper is well written and structured; however, some minor amendments are required based on the following comments and suggestions.
L43 check the reference style. This is the only reference showing the year of publication.
L134 Could not find the mixing procedure for AW? How was the water added into the aggregates?
L146-149 The last two steps are confusing? From Table 2 we understood that there are 3 formulations AD, AW, and CW. So Why only the mixture was sieved under water? what about AD and AW? I assume both went through attrition and should be washed and sieved? It might be better to update section 2.3. to clarity the procedure for the three formulations. I would not call AD as a mixture.
L149 Were the dry coarse aggregates weighed before mixing?
L153 Define 'init' as initial.
L157 check the reference style pls.
L163-170 Is this a standard test? and what is the purpose of it, please elaborate?
L191 If the dry aggregate is AD please make it clear as AD?
L197 Does wet formulation mean AW? Please use the acronym from Table 2 in the discussion.
L207 Is this mean only aggregate particle were simulated but the model will reflect the cohesiveness?
L312-314 The particle motion behaviour of CC/500 looks different between the two models! pls clarify which cases have similar behaviour?
L315 please refer to Figure 5. However, it is difficult to see the particles closer to the walls compared to Figure 6. This could be clearer if there is a top view perspective such as figure 7.
L362-364 Any possible explanation why the CO-current had more attrition that CC?
L358-393 Is there any approach to present the error values between the experimental and simulation results?
L468 Please demonstrate why this work is important and how it can benefit the existing/future research and industry.
………………. End of comments …………………………….
Author Response
Response to Reviewer 2 Comments
The authors of this manuscript greatly appreciate the editor of the journal and the reviewers for the valuable feedback and comments provided. The authors have carefully considered all comments and suggestions forwarded by the reviewers. Accordingly, they are addressed in the revised manuscript.
The reviewers’ comments are listed below. Author replies are given under each comment (in red).
Point 1: L43 check the reference style. This is the only reference showing the year of publication.
Response 1: Modified as requested.
Point 2: L134 Could not find the mixing procedure for AW? How was the water added into the aggregates?
Response 2: The mixing procedure is the same for both AD and AW, described in section 2.3. A new line was included in the experimental procedure in order to clarify this step. We have also further explained the way to add the water into aggregates.
Point 3: L146-149 The last two steps are confusing? From Table 2 we understood that there are 3 formulations AD, AW, and CW. So Why only the mixture was sieved under water? what about AD and AW? I assume both went through attrition and should be washed and sieved? It might be better to update section 2.3. to clarity the procedure for the three formulations. I would not call AD as a mixture.
Response 3: We agree with the reviewer, the procedure was a little confusing since it was employed the term “mixture” in wrong way. Both AD, AW and CW were washed and sieved on a 2.5 mm sieve. We have modified the experimental procedure in order to clarify it.
Point 4: L149 Were the dry coarse aggregates weighed before mixing?
Response 4: Yes, they were. These represented the Minitial values used in computing the attrition results in the mixing tests.
Point 5: L153 Define 'init' as initial.
Response 5: Defined, as suggested.
Point 6: L157 check the reference style pls.
Response 6: Modified as requested.
Point 7: L163-170 Is this a standard test? and what is the purpose of it, please elaborate.
Response 7: This is not a standard test. A statement has been included in this section clarifying the purpose of the test, which was to provide information for calibrating the single-particle attrition model.
Point 8: L197 Does wet formulation mean AW? Please use the acronym from Table 2 in the discussion.
Response 8: These are AW and CW. This clarification was included in the text.
Point 9: L207 Is this mean only aggregate particle were simulated but the model will reflect the cohesiveness?
Response 9: This is correct.
Point 10: L312-314 The particle motion behaviour of CC/500 looks different between the two models! pls clarify which cases have similar behaviour?
Response 10: The reviewer is correct. The statement has been removed from the text.
Point 11: L315 please refer to Figure 5. However, it is difficult to see the particles closer to the walls compared to Figure 6. This could be clearer if there is a top view perspective such as figure 7.
Response 11: The text was modified, including reference to Figure 5. We refrained from including a top view of Figure 5 since the manuscript is relatively long already.
Point 12: L362-364 Any possible explanation why the CO-current had more attrition that CC?
Response 12: Actually this result is consistent with the collision energy spectra presented in Figures 8 and 9, which show that the frequencies of co-current collisions supersede those for the counter-current mode, for similar impact velocities. Reasons for this relatively surprising result are not evident at this time.
Point 13: L358-393 Is there any approach to present the error values between the experimental and simulation results?
Response 13: This is certainly a valid suggestion. We included comments about it in the referred section, as well as in the abstract and conclusions. By the way, the average mean error was estimated as 26.4%.
Point 14: L468 Please demonstrate why this work is important and how it can benefit the existing/future research and industry.
Response 14: A statement has been included mentioning that such knowledge can be used to design systems that maximize mixing while minimizing mechanical degradation.

Reviewer 3 Report
Dear Authors,
This paper shows an interesting hot topic that is essential to understand from the recycled concrete aggregate. I see the structure is well addressed and the paper, in general, showing a good agreement of the journal instructions. Thanks for your efforts.
I have the following comments on the paper before considering it for publication. I would like that, if possible, authors can clarify and address the comments below;
Comments:
1- It would be good if the authors show the development of this approach and compared it to previous studies? Why your approach can be significant to achieve better results?
2- Figure 8 and 9 need to have a better scale which can show different CO and CC configurations. Also please try to describe in better approaches and explain differences of results.
3- Please, re-write the conclusion and reflect on your results. Besides, explain the possible advantages and disadvantages of this approach?
4- Abstract is a bit confusing, as the reader will not be fully understanding why this approach is contributed? Why not others and what can be replaced from SoA.
5- Mostly the procedure is considering experimental works and numerical modelling, please indicate the limitation of your study and also consider some NDT methodologies which can take place to explore placed recycled concrete aggregate or possible to be considered for literature review, if still, it is not possible, please give some explanations?
Please consider re-review according to the above-mentioned comments, I suggest a major revision of the paper.
Looking forward to receiving the revised version.

Author Response
Response to Reviewer 3 Comments
The authors of this manuscript greatly appreciate the editor of the journal and the reviewers for the valuable feedback and comments provided. The authors have carefully considered all comments and suggestions forwarded by the reviewers. Accordingly, they are addressed in the revised manuscript.
The reviewers’ comments are listed below. Author replies are given under each comment (in red).
Point 1: It would be good if the authors show the development of this approach and compared it to previous studies? Why your approach can be significant to achieve better results?
Response 1: Actually, the approach proposed in the present work is novel and has not been previously applied to investigate degradation of particulate materials during mixing. One important part of the model, which is related to the attrition model, has been proposed in an earlier study [17] and has been applied for the first time to materials other than iron ore pellets.
Point 2: Figure 8 and 9 need to have a better scale which can show different CO and CC configurations. Also please try to describe in better approaches and explain differences of results.
Response 2: As requested, limits of both the x and the y axes have been changed in order to improved clarity. Discussion of Figures 8 and 9 has been revised, in order to more clearly reflect the content of the figures.
Point 3: Please, re-write the conclusion and reflect on your results. Besides, explain the possible advantages and disadvantages of this approach?
Response 3: Most paragraphs have been rewritten in other the better reflect the results.
Point 4: Abstract is a bit confusing, as the reader will not be fully understanding why this approach is contributed? Why not others and what can be replaced from SoA.
Response 4: The abstract has been revised so as to clarify the contribution from the present approach.
Point 5: Mostly the procedure is considering experimental works and numerical modelling, please indicate the limitation of your study and also consider some NDT methodologies which can take place to explore placed recycled concrete aggregate or possible to be considered for literature review, if still, it is not possible, please give some explanations?
Response 5: Indications of the study are highlighted. However, we believe examination of NDT technologies will change the line of thought of the introduction.
In this revised version of the manuscript we revised the entire text for grammar, style as well as adherence to the standard of the journal.

Round 2
Reviewer 3 Report
Dear authors,
Thank you for addressing all comments and the paper is ready for publicaton.
Bests,